# Myosin Isoform-Dependent Effect of Omecamtiv Mecarbil on the Regulation of Force Generation in Human Cardiac Muscle

**DOI:** 10.3390/ijms25189784

**Published:** 2024-09-10

**Authors:** Beatrice Scellini, Nicoletta Piroddi, Marica Dente, J. Manuel Pioner, Cecilia Ferrantini, Corrado Poggesi, Chiara Tesi

**Affiliations:** 1Department of Experimental and Clinical Medicine, University of Florence, 50134 Florence, Italy; beatrice.scellini@unifi.it (B.S.); nicoletta.piroddi@unifi.it (N.P.); marica.dente@unifi.it (M.D.); cecilia.ferrantini@unifi.it (C.F.); corrado.poggesi@unifi.it (C.P.); 2Department of Biology, University of Florence, 50134 Florence, Italy; josemanuel.pioner@unifi.it

**Keywords:** cardiac muscle, omecamtiv mecarbil, acto-myosin interactions, inorganic phosphate, myosin isoforms, contraction regulation, cross-bridge, human cardiac muscle

## Abstract

Omecamtiv mecarbil (OM) is a small molecule that has been shown to improve the function of the slow human ventricular myosin (MyHC) motor through a complex perturbation of the thin/thick filament regulatory state of the sarcomere mediated by binding to myosin allosteric sites coupled to inorganic phosphate (Pi) release. Here, myofibrils from samples of human left ventricle (β-slow MyHC-7) and left atrium (α-fast MyHC-6) from healthy donors were used to study the differential effects of μmolar [OM] on isometric force in relaxing conditions (pCa 9.0) and at maximal (pCa 4.5) or half-maximal (pCa 5.75) calcium activation, both under control conditions (15 °C; equimolar DMSO; contaminant inorganic phosphate [Pi] ~170 μM) and in the presence of 5 mM [Pi]. The activation state and OM concentration within the contractile lattice were rapidly altered by fast solution switching, demonstrating that the effect of OM was rapid and fully reversible with dose-dependent and myosin isoform-dependent features. In MyHC-7 ventricular myofibrils, OM increased submaximal and maximal Ca^2+^-activated isometric force with a complex dose-dependent effect peaking (40% increase) at 0.5 μM, whereas in MyHC-6 atrial myofibrils, it had no effect or—at concentrations above 5 µM—decreased the maximum Ca^2+^-activated force. In both ventricular and atrial myofibrils, OM strongly depressed the kinetics of force development and relaxation up to 90% at 10 μM [OM] and reduced the inhibition of force by inorganic phosphate. Interestingly, in the ventricle, but not in the atrium, OM induced a large dose-dependent Ca^2+^-independent force development and an increase in basal ATPase that were abolished by the presence of millimolar inorganic phosphate, consistent with the hypothesis that the widely reported Ca^2+^-sensitising effect of OM may be coupled to a change in the state of the thick filaments that resembles the on–off regulation of thin filaments by Ca^2+^. The complexity of this scenario may help to understand the disappointing results of clinical trials testing OM as inotropic support in systolic heart failure compared with currently available inotropic drugs that alter the calcium signalling cascade.

## 1. Introduction

Over the past 15 years, the ongoing search for new cardiotonic agents for the treatment of decompensated heart failure (HF) has evolved from membrane targets to the attractive concept of pharmacological agents targeting the myocardial contractile protein machinery. Omecamtiv mecarbil (OM) is a cardiac myosin activator identified in a small molecule screen using the thin filament-activated ATPase activity of β-cardiac myosin [1]. OM was developed as a first-in-class drug for the treatment of heart failure with reduced ejection fraction (HFrEF) and is one of the myosin modulators that represent the new frontier in cardiovascular medicine [2,3]. Several clinical trials (ATOMIC-HF, COSMIC-HF, GALACTIC-HF, METEORIC-HF) have been conducted to test its efficacy in HFrEF patients, avoiding the side effects of most inotropic drugs ([4,5,6,7], for review [8,9]). Although clinical trial results showed good tolerability and positive outcomes compared with placebos in reducing cardiovascular death or first HF events [6], the drug has not yet been approved by the FDA for human treatment, partly because of the increased mortality seen in patients with concomitant atrial fibrillation (AF) [10] and because of the many unresolved concerns about its use [9], including the possible side effects on slow skeletal muscle [11]. The hope that OM will be able to benefit a selected population of high-risk HFrEF still awaits a deeper understanding of its pharmacological action at the clinical level, which will go hand in hand with the clarification of the many obscure points regarding the molecular mechanism of its action as a myosin modulator.

OM is known to act on the myosin motor by binding to an allosteric site that stabilises the lever arm in a primed position and then increases the number of recruitable myosin heads for each cardiac cycle [12]. The mechanism of action of OM as a cardiac activator is purely sarcomeric as it does not affect Ca^2+^ transients [1,13], but its effect is strongly dependent on the level of activation, being higher at lower [Ca^2+^] [1,14], with important differences in different (animal and human) experimental models [14,15]. Much remains unclear about the mechanism of action of the drug, in particular, the apparent increase in Ca^2+^ sensitivity of OM, which suggests a complex mechanistic basis of action involving perturbation of the thin/thick filament regulatory state of the sarcomere mediated by binding to myosin allosteric sites coupled to inorganic phosphate (Pi) release [16]. Investigation of the mechanism of action of OM is then a key tool to perturb and gain insight into the regulation of force generation in striated muscle based on a “dual-filament regulatory mechanism” that positively couples the Ca^2+^-dependent (thin filament) and stress-dependent (thick filament) activation mechanisms [17]. By increasing the number of recruitable heads in the cardiac thick filament, OM would enhance myosin-sensing in the thin filament with beneficial consequences for cardiac function at the sarcomere level. The effect of OM is then expected to be critically tuned by differences in the interplay between the distribution states of the actomyosin ATPase cycle [18,19] in relation to actin affinity and the strong cross-bridge formation that sustains the thin filament regulatory “on” state [20] in the presence of different myosin motor isoforms. This hypothesis is supported by recent results obtained in the presence of α-fast MyH-6 (atrial) or β-slow MyH-7 (ventricular) myosin heavy chain isoform expression [19], resulting in greater prolongation of cross-bridge attachment time [21] and greater drug-induced Ca^2+^ sensitisation [22] in the ventricle than in the atrium, i.e., in association with the predominance of slow motors. Understanding the molecular mechanism of drug action would then benefit from elucidating the central role of myosin isoform dependence in OM effects, as explored in the human heart, which shows a sharp difference in myosin isoform expression in young and healthy subjects, with the atria predominantly expressing α-fast MyH-6 and the ventricles expressing almost 100% β-slow MyH-7. 

To this end, myofibrils from samples of human donor left ventricle and left atrium were used here to study the differential effects of micromolar concentrations of OM on isometric force under relaxed (pCa 9) and maximal (pCa 4.5) or half-maximal (pCa 5.75) Ca^2+^ activation, both under control conditions (15 °C, equimolar DMSO, contaminant [Pi] ~170 μM) and in the presence of mM [Pi]. The main objectives of this study were to use OM as a sensitive tool to detect key differences in the cross-bridge mechanical cycle in the presence of predominant α-fast or β-slow myosin heavy chain isoforms in relation to the regulatory state of ventricular and atrial sarcomeres and to characterise the interplay between OM and Pi in modulating the mechanical performance of human myocardium. Understanding the complex mechanism of action of OM on cardiac sarcomeres in relation to regional gene expression patterns and metabolic conditions may also help explain disappointing results in clinical trials and inform future drug development strategies for the treatment of HFrEF based on small molecules acting as myosin modulators.

## 2. Results

### 2.1. OM Induces a Rapid and Fully Reversible Modulation of Force Generation in Human Atrial and Ventricular Myofibrils

The effect of OM on the contractile properties of human myocardium was studied in thin bundles of myofibrils (1.5–4.0 µm diameter) isolated from samples (see Section 4) of human left atrial (HA) or ventricular (HV) muscle and subjected to activation/relaxation cycles at 15 °C using our previously described rapid solution-switching protocols [23,24]. Atrial and ventricular myofibrils were characterised for MyHC composition (α-fast MyHC-6 or β-slow MyHC-7) by 8% SDS-PAGE (Figure 1A,B), HV being 96% MyHC-7 vs. MyHC-6 and HA being 83% MyHC-6 vs. MyHC-7 (Figure 1C). HV and HA myofibrils were mounted in relaxing solution of pCa 9.0 and optimal sarcomere length under control conditions or in the presence of selected [OM] from 0.05 to 10 μM (15 °C). Unless otherwise stated, all solutions had a contaminant inorganic phosphate concentration ([Pi]) of ∼170 μM [25] and were normalised to DMSO, which was used as the solvent for OM. Maximal force generation was elicited by rapidly switching the solution to maximal (pCa 4.5) or submaximal (pCa 5.75) Ca^2+^-activating conditions using two different experimental protocols. In the first protocol, the effect of OM on isometric force was measured in the same myofibril from the amplitude of an “OM jump” (see Section 4), i.e., the myofibril was first activated at pCa 4.5 or 5.75 under control conditions and, once a steady plateau of isometric force was reached, switched to the same activating solution containing a given [OM] and then back to the control solution with no OM (Figure 2A for HV and Figure 2C for HA). As the effect of OM on force development by HV and HA myofibrils was found to be very rapid and fully reversible, the jump protocol provided an internal control for estimating the effect of ligand concentration on contractile mechanics with high resolution, reducing the effect of inter-sample variability and rundown artefacts [25]. In the second protocol, two different batches of HV or HA myofibrils were tested for maximal (pCa 4.5) or submaximal (pCa 5.75) isometric force development in contraction–relaxation cycles under control conditions or in the presence of selected concentrations of OM in both relaxing and activating solutions (Figure 3A,B for HV and Figure 3E for HA). In both protocols, ligand concentrations in the myofibril lattice were assumed to be equal to those of the perfusing solution continuously flowing through the preparation. The resting sarcomere lengths of the myofibrils used in the experiments were 2.09 ± 0.01 µm (*n* = 144) for HV and 2.13 ± 0.01 µm (*n* = 126) for HA myofibrils.

The mean maximum isometric tension of all myofibrils tested with the two protocols under control conditions (in the presence of DMSO) was 82 ± 5 (*n* = 55) and 82 ± 6 (*n* = 49) mN mm^−2^ for HV and HA myofibrils, respectively. These values are similar to those previously reported for the same myofibril system at 15 °C in the absence of DMSO [23] and confirm the absence of any significant effect of DMSO up to 0.5% on cardiac sarcomeres [24]. As previously reported [19], the prevalence of fast or slow myosin isoforms did not significantly affect the maximal force developed but strongly modulated the rate of force development and relaxation (see Table 1). A limited number of myofibrils were also tested at a submaximal Ca^2+^ activation of pCa 5.75, which is approximately pCa_50_ for maximal isometric force development under control conditions [23]. In all conditions tested, the effect of OM proved to be very rapid and fully reversible in both HV and HA myofibrils.

### 2.2. HV Myofibrils: OM Increases Tension and Decreases the Kinetics of Force Generation and Relaxation Simultaneously Perturbing the Regulation State of Ventricular β-Slow MyHC-7 Sarcomeres

As shown in Figure 2B, OM had a complex effect on maximal (pCa 4.5, black dots) and submaximal (pCa 5.75, grey dots) Ca^2+^-activated tension of HV myofibrils, increasing it up to approximately 40% for 0.5 μM OM at pCa 4.5 and for 1 μM OM at pCa 5.75. Higher concentrations of OM resulted in no effect or a small decrease in force. The force potentiation for 0.5 μM [OM] jumps in the maximally activating solution (1.40 ± 0.11, *n* = 8) was not affected by the presence of 5 mM [Pi] (1.45 ± 0.05, *n* = 5; see also Figure 3D). At the same time, OM dramatically affected the kinetics of force development as measured by *k*_TR_ in a dose-dependent manner, decreasing it by approximately 50% of the control value at a concentration of 0.1 μM [OM], for both full (black triangles) and submaximal (grey triangles) Ca^2+^-activated conditions. Furthermore, the effect of [OM] on the contractile properties of HV myofibrils was fully characterised using the activation–relaxation protocol in the presence of different OM concentrations added to both relaxing and activating solutions in the presence of the contaminant (Figure 3A) or 5 mM inorganic phosphate (Figure 3B). The results of these experiments in different myofibril batches confirmed the effect of the drug on tension detected by jump-coupled experiments. Moreover, since this protocol involves exposure of myofibrils to OM in a relaxing solution, it was possible to demonstrate that OM alone was able to switch on the contractile apparatus of ventricular sarcomeres in the absence of Ca^2+^ in a dose-dependent manner. As shown in Figure 3C, at a drug concentration of 10 μM, HV myofibrils developed a Ca^2+^-independent force (open symbols) that reached a value comparable to active force generation (filled symbols). Notably, the rate of OM-induced Ca^2+^-independent force development *k*_OM_ was found to be independent of ligand concentration with a value of 0.028 ± 0.003 s^−1^ (*n* = 6). The addition of 5 mM [Pi] to the experimental solutions completely abolished the Ca^2+^-independent force development induced by OM (Figure 3B,D), whereas the force potentiation on Ca^2+^ activation was fully preserved, suggesting a close relationship between OM and Pi binding sites on the myosin head. Measurements from the activation–relaxation protocol of HV myofibrils in the presence of different concentrations of OM confirmed the dramatic and dose-dependent slowing effect of the drug on the kinetics of force development, also when measured by *k*_ACT_, (Figure 4A). Furthermore, OM was also shown to slow the kinetics of force relaxation by prolonging the duration of the slow phase (Figure 4B) and decreasing both *slow k*_REL_ (Figure 4C) and *fast k*_REL_ (Figure 4D), with a 50% decrease at approximately 0.1 μM [OM]. Notably, all the measured rates for the kinetics of force development and relaxation tended to a similar asymptotic value at 10 μM [OM] (*k*_ACT_ 0.021 ± 0.002; *slow k*_REL_ 0.024 ± 0.003 s^−1^, *fast k*_REL_ 0.027 ± 0.008 s^−1^, *n* = 4), which roughly corresponded to the rate of OM-induced Ca^2+^-independent force development *k*_OM_.

### 2.3. HA Myofibrils: OM Increases Submaximal Tension and Decreases the Kinetics of Force Generation and Relaxation with Little Effect on the Regulatory State of Atrial α-Fast MyHC-6 Sarcomeres

When tested on HA myofibrils in jump experiments (Figure 2D), OM had similar effects on the rate of force redevelopment *k*_TR_ as observed in HV myofibrils, but the potentiation of force was only evident at submaximal pCa 5.75 (grey dots) and not at full Ca^2+^ activation (black dots), where a significant decrease in force was present for [OM] higher than 5 μM. As in the case of HV myofibrils, the *k*_TR_ of HA myofibrils was also dramatically and dose-dependently slowed by OM, being 40–50% of the maximal value at 0.1 µM [OM], at both maximal (black triangles) and submaximal (grey triangles) levels of Ca^2+^-activation. Activation–relaxation protocols performed on HA myofibrils in the presence of different concentrations of OM added to both the relaxing and activating solutions confirmed the effect on force development; however, in contrast to what was observed in HV myofibrils, [OM] < 1 µM did not induce significant Ca^2+^-independent force development (Figure 3E). Only high [OM]s in relaxing solution (1–10 µM) were able to induce a small amount of Ca^2+^-independent force in HA sarcomeres, which was still less than 10% of the maximal Ca^2+^-activated force (Figure 3F, open symbols). In HA myofibrils, the rate of OM-induced Ca^2+^-independent force development *k*_OM_ was found to be independent of ligand concentration with a value of 0.13 ± 0.03 s^−1^ (*n* = 4). It is worth noting that this rate is approximately 5 times faster compared with HV myofibrils. Also, in atrial sarcomeres, the presence of mM [Pi] in the relaxing solution abolished the Ca^2+^-independent force in the presence of OM. The dose-dependent slowing of the activation and relaxation kinetics of HA myofibrils in the presence of OM was confirmed by the activation–relaxation protocol (Figure 4 E-H). As with HV myofibrils, the effect was dose-dependent (50% decrease in *k*_ACT_, *slow k*_REL_ and *fast k*_REL_ at approximately 0.1 μM [OM]). Also, in HA myofibrils, the asymptotic value for the inhibition of the kinetics of active force development (*k*_ACT_ 0.11 ± 0.01 s^−1^) and relaxation (*slow k*_REL_ 0.06 ± 0.001 s^−1^ and *fast k*_REL_ 0.39 ± 0.012 s^−1^, *n* = 4) roughly corresponded to the rate of Ca^2+^-independent force development *k*_OM_.

### 2.4. OM Reduces the Inhibition of Force by Pi in the Low [Pi] Range

To investigate the inter-relationship between OM and Pi action in the modulation of active force development by MyHC-7 or MyHC-6 myosin isoforms, we characterised the relationship between maximal Ca^2+^-activated force and [Pi] concentration in the range of 0.5–70 mM in HV (Figure 5A) and HA (Figure 5B) myofibrils, first in control (Ctrl) conditions (black symbols) and then in the presence of 0.5 μM [OM] (red symbols). Hyperbolic fits of the Ctrl HV and HA force/[ Pi] relationships yielded similar Pi_50_, values, namely, 0.83 ± 0.08 mM and 2.12 ± 0.73 mM for HV and HA myofibrils, respectively. The maximum amplitude of force depression by Pi was smaller in HV (0.83 ± 0.08) compared with HA myofibrils (0.29 ± 0.01). The presence of 0.5 μM [OM] in the activating solution had a small but significant effect on both HV and HA force/[Pi] relations, but only in the low [Pi] range did the drug reduce the inhibition of force by Pi. The effect was approximately maximal at 5 mM [Pi] and greater in HV myofibrils. Unfortunately, the steepness of the force/[Pi] relationships in the millimolar Pi range made it very difficult to give accurate quantitative estimates of the extent of the effect (20–40%). The effect of [OM] was not significant at [Pi] concentrations above 10 mM.

### 2.5. Effect of OM on the Resting Tension and ATPase of Skinned Strips of Human Ventricular Muscle

To gain further insight into the mechanism by which OM induces Ca^2+^-independent force development in MyHC-7 sarcomeres, we tested the effect of OM on resting tension and ATPase in permeabilised human ventricular strips at 25 °C. As shown in Figure 6 (A and C, left panel), the presence of 5 µM [OM] induced a perturbation of the regulatory state in pCa 9.0, resulting in the development of a large amount of Ca^2+^-independent force over resting tension (from 0.55 ± 0.13 to 7.19 ± 1.3 mN mm^−2^, *n* = 6). In addition, 5 µM [OM] induced a 2-fold increase in resting ATPase activity (B and C, right panel), which was increased from 21 ± 5 to 42 ± 14 pmol μL^−1^ s^−1^ (corresponding to turnover rates of 0.13 and 0.26 s^−1^, respectively, when normalised to the myosin concentration of ventricular muscle [26]). Similar to what was observed in cardiac myofibrils, the presence of mM [Pi] in the relaxing solution completely abolished the effect of OM by rapidly returning the resting tension to the control level (Figure 6A, time window marked by the arrow).

## 3. Discussion

This study, based on mechanical measurements in human cardiac myofibrils, shows that OM exerts a strong and myosin isoform-dependent influence on the mechanism of force generation and its regulation. Under the conditions of the present study (15 °C, ~170 µM contaminant [Pi]), OM exerts a complex, dose-dependent and myosin isoform-dependent influence on both Ca^2+^-independent and Ca^2+^-activated isometric forces (at both maximal and submaximal activation). The effects of OM are modulated by the presence of inorganic phosphate. The rapid jump protocol also shows that the sarcomeric effects of OM are rapid and fully reversible. The complexity of this scenario may explain, at least in part, the disparate and sometimes even contradictory results reported in the extensive body of literature devoted to the study of this molecule over the last decade using different models of striated muscle, mainly in the presence of unidentified inorganic phosphate contaminants [2,3,9]. A better understanding of the differential molecular aspects of OM action in relation to dose, regional distribution of fast and slow cardiac myosin isoforms and ATPase reaction environment may help to understand the disappointing results of clinical trials testing OM as inotropic support in systolic heart failure compared with currently available inotropic drugs that alter the Ca^2+^ signalling cascade [27].

The present study reports that [OM] in the sub-micromolar range significantly increases Ca^2+^-activated force in both human atrial and ventricular myofibrils at submaximal Ca^2+^-activation, in agreement with the extensive body of evidence accumulated over the years in the myocardium and slow skeletal muscle from animal models [22,28,29,30,31] and in human ventricular samples [15,28]. While OM also increased the maximum Ca^2+^-activated force of slow MyHC-7 ventricular sarcomeres, it had no effect (or decreased at concentrations higher than 5 µM) on the maximum activated force of fast MyHC-6 atrial sarcomeres. This specific finding adds to several previous observations that OM effects on maximal isometric force vary with experimental models and conditions. For instance, at full Ca^2+^ activation force has been reported to increase (our results in HV, and [22] in porcine heart at 15 °C), remain unchanged ([15,28] and our results in HA), or decrease ([14,29,32] and [16] for rabbit soleus). These results highlight the need for a defined experimental milieu and caution should be exercised when reporting the effect of OM on contractile properties.

On the other hand, our results are consistent with previous observations showing that OM exerts a pronounced inhibitory effect on the kinetics of force generation and relaxation [1,15,29]. Irrespective of the presence of fast or slow myosin isoforms within the sarcomere, OM exerted a pronounced inhibitory effect on the force development kinetics of HV and HA myofibrils, with a maximum reduction of up to 90% observed at 10 µM [OM]. OM also significantly affected the kinetics of force relaxation in HV and HA myofibrils, prolonging the slow phase and reducing the rate of both fast and slow relaxation. This finding is consistent with previous observations in permeabilised human ventricular cardiomyocytes [32] and intact atrial trabeculae [33]. It is noteworthy that as [OM] was increased in the mM concentration range, the rates of force development and relaxation observed in HV and HA myofibrils showed convergence to the same extremely low asymptotic level. This level was approximately 5 times faster in fast MyHC-6 versus slow MyHC-7 beta human cardiac sarcomeres.

In addition, these results support the kinetic effect of OM through an increased duty ratio that contributes to the formation of long-lived, strongly bound acto-myosin states that configure cross-bridges on either the conventional or non-conventional parallel force-generating cycle [15,16,34]. These OM motors, with a stabilised configuration in the pre-powerstroke state [12,29,35,36,37,38] and long attachment time [36,39], are able to induce cooperative activation of the thin filaments resulting in an increase in Ca^2+^ sensitivity [14,29,40]. Furthermore, our results show that the effect on the duty ratio is independent of the turnover rate of the myosin motor isoforms, at least for the fast and slow cardiac MyHC isoforms, since the estimated dissociation constant (*Kd)* for the inhibition of *k*_ACT_ was similar in HV and HA myofibrils (0.08 ± 0.01 µM and 0.11 ± 0.02 µM, respectively). The estimated *Kd* values for the inhibition of *slow k*_REL_ were also the same in HV and HA myofibrils (0.08 ± 0.05 and 0.06 ± 0.01 µM, respectively). These values are in the range of the *EC_50_* values reported for the inhibition of the working stroke in cardiac MyHC-7 myosin from in vitro motility or steady-state actin-activated ATPase (0.1–0.5 µM) [15,34,36].

Another striking isoform-dependent effect of OM concerns the ability to activate Ca^2+^-independent force, which implies the presence of important differences in the ability of MyHC-6 and MyHC-7 myosin heads with OM bound to bind to actin and turn on the thin filaments. This is particularly evident in ventricular myofibrils where the effect appears at sub-micromolar concentrations of the ligand and approaches the maximum Ca^2+^-activated tension value in the presence of 10 µM [OM]. The possibility that these effects observed in isolated HV myofibrils at 15 °C could be due to a partial loss of the thick-filament regulatory mechanism associated with our experimental conditions was ruled out by experiments in skinned human LV strips (Figure 6). These experiments confirmed that 5 µM OM was able to induce Ca^2+^-independent force development also at 25 °C and in the presence of 5% dextran, conditions that have been reported [40] to increase the order state of heads on the thick filament backbone. The same experiments also showed an increase in cardiac strip ATPase in the virtual absence of activating Ca^2+^, confirming the Ca^2+^-independent activation of sarcomere contraction. The results obtained in HV demembranated strips at 25 °C also allowed for the measurement of the parallel increase in the resting ATPase in the presence of OM, as expected from the activation of contraction by OM in the virtual absence of Ca^2+^. Interestingly, MyHC-6 OM-bound motor heads show a reduced ability to turn on thin filaments, as a small force (less than 10%) was generated by atrial myofibrils only at very high [OM] (10 µM). The rate of Ca^2+^-independent force development *k*_OM_ was about 3–4 times faster in HA (~0.2 s^−1^) than in HV (~0.04 s^−1^) myofibrils, and, in the latter case, the rate was also shown to be dose-independent. In HV and HA, these rates correspond to the asymptotic values of the dose-dependent inhibition of the kinetics of force development and relaxation in the presence of OM. Interestingly, the absolute values of these rates are in the same range as the ATPase turnover rates of MyHC-7 and MyHC-6 motors measured in coupled assays under relaxing conditions at room temperature in demembranated human ventricular and atrial myocardial strips (recalculated from [41]; 2–0.4 s^−1^ and 0.04–0.05 s^−1^) or from alpha fast mouse ventricle or beta slow minipig ventricle (0.11 s^−1^, Dente et al., unpublished ). On this basis, we hypothesise that following Pi release and M-ADP state formation (according to the specific turnover rate of each isoform), OM binds to myosin heads, forming a conformation of higher actin affinity, capable of cooperatively binding to the thin filament and displacing the tropomyosin–troponin complex [42]. Thin filament activation by OM-bound myosin heads was already postulated [14] and is considered the mechanistic reason for the increase in Ca^2+^ sensitivity and the decrease in cooperativity of pCa force curves in the presence of OM [15,16,30]. This OM-induced thin filament activation is shown here to be isoform-dependent, being much stronger in slower myosin. This can be explained by the same hypothesis raised to explain the greater Ca^2+^ sensitising effect of µM [OM] in porcine ventricles compared with atria [22] based on in vitro motility and optical trap studies of isolated fast and slow cardiac motors [21], showing that OM induces a prolongation of cross-bridge attachment time that is greater in ventricular than in atrial myosin, allosterically promoting strong cross-bridge formation and cooperative thin filament activation. The Ca^2+^-independent nature of this mechanism has been hypothesised based on the absence of significant changes in the orientation of fluorescent probes on Troponin C signalling thin filament structural states associated with probes [14] and from the modulation of MgADP-induced contractions observed in cardiac muscle [22].

Ca^2+^-independent force generation in the presence of OM is also controversial, which has been reported in MyHC-7 cardiac muscle of animal models [14,29] but has not been demonstrated or addressed in other cases [16,22,28,30]. Notably, permeabilised human LV cardiomyocytes in the presence of 1 µM [OM] (i.e., the drug serum concentration used in clinical trials [43]) showed a reduction in diastolic sarcomere length and an increase in passive stiffness as a consequence of the increased Ca^2+^ sensitivity and Ca^2+^-independent thin filament activation. That study linked this effect to the in vivo impairment of ventricular function observed by the authors in animal models [32]. This, together with the slowing of contractile kinetics led the authors to conclude that OM designed to improve systolic function also induces diastolic dysfunction, making the window for its therapeutic use rather narrow. Our results in HV and HA myofibrils help to clarify this issue and the discrepancy among observations present in the literature, as the development of Ca^2+^-independent force is markedly isoform dependent and is abolished by Pi concentrations (1–2 mM) that are in the range of those physiologically present in cardiac tissue [42]. The present results are then consistent with OM acting through a complex perturbation of the thin/thick regulatory state of the sarcomere mediated by binding to allosteric sites coupled to Pi release. The close interplay between OM and Pi in modulating cardiac mechanical performance had been previously suggested based on observations in rabbit soleus muscle expressing MyHC-7 [16] indicating a reduction in force inhibition operated by Pi in the low 1–2 mM [Pi] range in the presence of OM. This result was fully confirmed here for both MyHC-7 and MyHC-6 isoforms expressed in the context of human cardiac sarcomeric proteins, where OM was shown to reduce the inhibition of force exerted by Pi, but only slightly and only in the same low [Pi] range, in the presence of no significant difference in the *K_d_* values for the force/[Pi] relationships between HV and HA myofibrils.

In conclusion, OM is a sensitive tool that can be used to study the interplay between thin- and thick-filament mechanisms in the cooperative and allosteric interactions associated with the regulation of cardiac muscle by Ca^2+^ and the formation of strong cross-bridges. This is particularly true in the absence of Ca^2+^ and under conditions of low Pi, which favour the formation of strong cross-bridges, thereby enhancing the regulatory contribution of motor heads strongly bound to the actin filament. OM is also an additional sensitive tool that can be used to detect key differences in the distribution of states of the actomyosin ATPase cycle of the atrial and slow ventricular MyHC isoforms in resting and activated conditions, resulting in the diastolic and systolic properties of cardiac function. Unfortunately, our results underline how important factors such as dose, myosin isoform composition, and Pi concentration can dramatically affect the results obtained from different experimental models and conditions. Age, gender, and pathology-related differences in the regional expression of human cardiac MHC genes [44] add a further layer of complexity that may contribute to the inconsistent results in experimental OM studies and clinical trials. This must be carefully considered by evaluating the many caveats raised about its therapeutic use as an off-target cardiac inotrope.

## 4. Methods

### 4.1. Preparation of Myofibrils from Human Cardiac Sample

Human cardiac myofibrils were prepared by homogenising frozen human left atrial and ventricular samples from healthy donors (one male, 60 years old, and one female, 24 years old, respectively) stored at −80 °C in the Da Vinci Biobank of the University of Florence. Experiments involving the use of human samples were approved by the local ethics committee (Comitato Etico Regionale per la Sperimentazione Clinica della Regione Toscana, 19337_bio, 13 May 2022). Thin strips dissected from cardiac samples were permeabilised overnight in an ice-cold relaxing solution added with 1% Triton™ X-100 (D-76185, Karlsruhe, Germany). Demembranated strips were then homogenised in relaxing solution to prepare myofibril suspensions [23]. The suspensions stored at 0–4 °C were stable and were used for up to 5 days. All solutions to which the myofibrils were exposed contained a cocktail of protease inhibitors including leupeptin (10 µM), pepstatin (5 µM), phenylmethylsulphonyl fluoride (200 µM), and E-64 (10 µM), as well as NaN_3_ (500 µM) and 500 µM dithiothreitol. [Ca^2+^] in the experimental solutions was expressed as pCa = −log[Ca^2+^].

### 4.2. Electrophoretic Assays

1-D SDS-polyacrylamide gel electrophoresis and densitometric analysis were used to determine the MyHC isoform composition (α-fast MyHC-6 or β-slow MyHC-7) in the cardiac myofibril suspensions according to previously described procedures [45,46]. These analyses used a Mini-PROTEAN 3 Cell system (Bio-Rad, Hercules, CA, USA) and hand-cast non-gradient denaturing gels loaded with cardiac myofibril samples solubilised at 95 °C for 3–5 min in Laemmli buffer solution. Stacking and separating gels consisted of 4% and 8% acrylamide (*wt*/*vol*), respectively, with a 50:1 ratio of acrylamide–N,N8-methylene-bis-acrylamide. Gels were run at 4 °C for 20 h at 70 V and then stained with 0.1% Coomassie Brilliant Blue R-250 (Merck Life Science, St. Louis, MO, USA) to visualise the resolved protein bands. Specifically, the thickness of the gel (1 mm instead of 0.75 mm to reduce resistance), the lower concentration of glycerol in the separating gel, and the lower voltage associated with a longer run time allowed us to achieve the resolution required for the separation of the two adult cardiac MyHC isoforms. The gels were digitised and analysed with UN-SCAN-IT gel 6.0 software (Silk Scientific, Inc., Orem, UT, USA), which allows the quantification of band intensity. Each band (α MyHC-6 or β MyHC-7) was expressed as a percentage of the total MyHC content.

### 4.3. Myofibril Experiments

Bundles of a few myofibrils (30–90 µm long and 1.5–4 µm wide) were mounted in a force recording apparatus as previously described [25]. Briefly, myofibrils were mounted horizontally between two glass microtools in a temperature-controlled chamber (15 °C) filled with a relaxing solution (pCa 9.0). One tool was connected to a length-control motor capable of producing rapid (<1 ms) changes in length. The second tool was a calibrated cantilever force probe (2–6 nm/nN; frequency response of 2–5 kHz). Force was measured from the deflection of the force probe image projected onto a split photodiode. The initial sarcomere length of the preparations was set just above the slack length. Myofibrils were activated and relaxed by rapidly moving the interface between two flowing streams of the activating (pCa 4.5) and relaxing (pCa 9.0) solutions across the preparation, under control conditions (Ctrl) or in the presence of selected OM concentrations in both solutions. The solution change was completed in <5 ms. The maximal developed force (P_0)_ was measured and normalised for the cross-sectional area of the preparation. The rate of force development (*k*_ACT_) and the rate of force redevelopment following a release–restretch protocol (*k*_TR_) [47], as well as (when present) the rate of calcium-independent force development in the presence of OM (*k*_OM_**),** were estimated from the time required to reach 50% of the steady-state isometric force. Total internal shortening during active force development was below 10%.

The rate constant of the early slow force decay (*slow k*_REL_) was estimated from the slope of the regression line fitted to the tension trace normalised to the active tension just before relaxation. The early slow force decay (linear phase of relaxation) is assumed to be the initial part of an exponential process which, if it continued throughout the relaxation transient, will bring the active force to zero with a rate constant equal to the initial slope of the force decay divided by the amplitude of the overall force decay. Experimental traces were not used to measure *slow k*_REL_ if the mechanical artefacts produced by the solution change did not allow for reproducible measurements (±10%) by two different investigators. The duration of the slow relaxation phase was estimated from the onset of the solution change signal. The rate constant for the final fast phase of tension decay (*fast k*_REL_) was estimated from a mono-exponential fit [48].

In the OM-jump experiments, both channels of the perfusion pipette were loaded with the activating solutions, one without OM (Ctrl) and the other one with OM. Myofibrils were activated by moving the interface between the relaxing solution in the experimental chamber and the flow of the Ctrl activating solution. Once a steady plateau of isometric force was reached, the perfusing flow was rapidly switched to the activating solution containing selected concentrations of OM and back. The constant level of force was measured during the drug exposure period as well as *k*_TR_ following the imposition of a release–restretch protocol.

### 4.4. Cardiac Strip Experiments

The experimental procedures, solutions, and equipment for the simultaneous measurement of active and resting tension and ATPase activity by an enzyme-coupled assay in human permeabilised strips from the left ventricle samples were as previously described [41]. Maximal isometric tension and ATP consumption were measured at 25 °C under resting conditions (pCa 9.0) or at saturating [Ca^2+^] (pCa 4.5) under Ctrl conditions or in the presence of 5 µM [OM]. The length of the preparations was adjusted to a sarcomere length of ∼2.2 µm [23]. ATPase activity was normalised to the volume of the muscle strip and to the myosin content in the myocardium [26].

### 4.5. Solutions for Mechanical Experiments

All activating and relaxing solutions were calculated as previously described at pH 7.0 and contained 10 or 1 mM of total EGTA (CaEGTA/EGTA ratio adjusted to give different pCa values in the range of 9.0–4.5), 5 mM MgATP, 1 mM free Mg^2+^, 10 mM 3-(N-morpholino) propanesulphonic acid, propionate, and sulphate to adjust the final solution to an ionic strength of 200 mM and a monovalent cation concentration of 155 mM. Creatine phosphate (10 mM) and creatine kinase (200 units mL^−1^) were added to all solutions to minimise changes in the concentration of MgATP and its hydrolysis products. Creatine kinase and creatine phosphate were not present in solutions containing 5 mM MgADP. Omecamtiv mecarbil (OM, APExBIO Technology LLC, Houston, TX, USA) was dissolved in DMSO to give a 10 mM stock solution. This solution was mixed with relaxing and activating solutions to give final test concentrations of 0.05–10 µM [OM], corresponding to 0.0005–0.1% DMSO (vol/vol). Control solutions were normalised for DMSO content. All chemicals and enzymes were purchased from Sigma-Aldrich (Merck Life Science, St. Louis, MO, USA).

### 4.6. Data Acquisition and Analysis

Force and length signals were continuously monitored throughout the experiments using commercial software and programs (LabVIEW, 2014, 14.0, National Instruments, Austin, TX, USA) modified for our use. The same signals were also recorded during the experimental protocols and used later for data analysis. Data measurements were made directly using commercial software (Origin 2018, 95E, OriginLab corp., Northampton, MA, USA) and a custom-written LabVIEW analysis program that converted the analogue signals to numerical values. Data were expressed and plotted as the mean ± SEM obtained from *n* myofibrils. Comparisons were made using a two-tailed Student’s *t* test. Differences between groups were considered statistically significant when *p* ≤ 0.05.

## Figures and Tables

**Figure 1 ijms-25-09784-f001:**
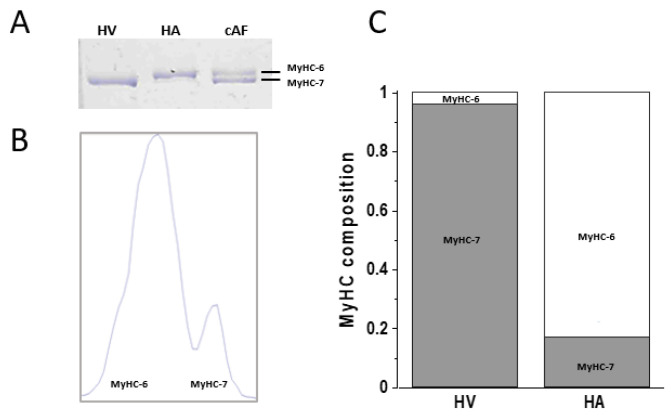
Myosin heavy chain isoform composition in human ventricular and atrial myofibrils. (**A**): Representative 8% SDS-PAGE of the MyHC isoforms in human cardiac samples. HV and HA: samples used to prepare the myofibrils used in this study; cAF: surgical sample from patients with chronic atrial fibrillation used as a reference standard for identification of α-fast MyHC-6 and β-slow MyHC-7 band positions. (**B**): Representative density profile of the MyHC isoforms in the HA sample (MyHC-6: 0.83; MyHC-7: 0.18). (**C**): Relative distribution of the MyHC isoforms in HV and HA myofibrils (HV: MyHC-6: 0.04 ± 0.01, β-slow MyHC-7: 0.96 ± 0.01, *n* = 6; HA: MyHC-6: 0.83 ± 0.01, β-slow MyHC-7: 0.17 ± 0.01, *n* = 21; means ± SEMs).

**Figure 2 ijms-25-09784-f002:**
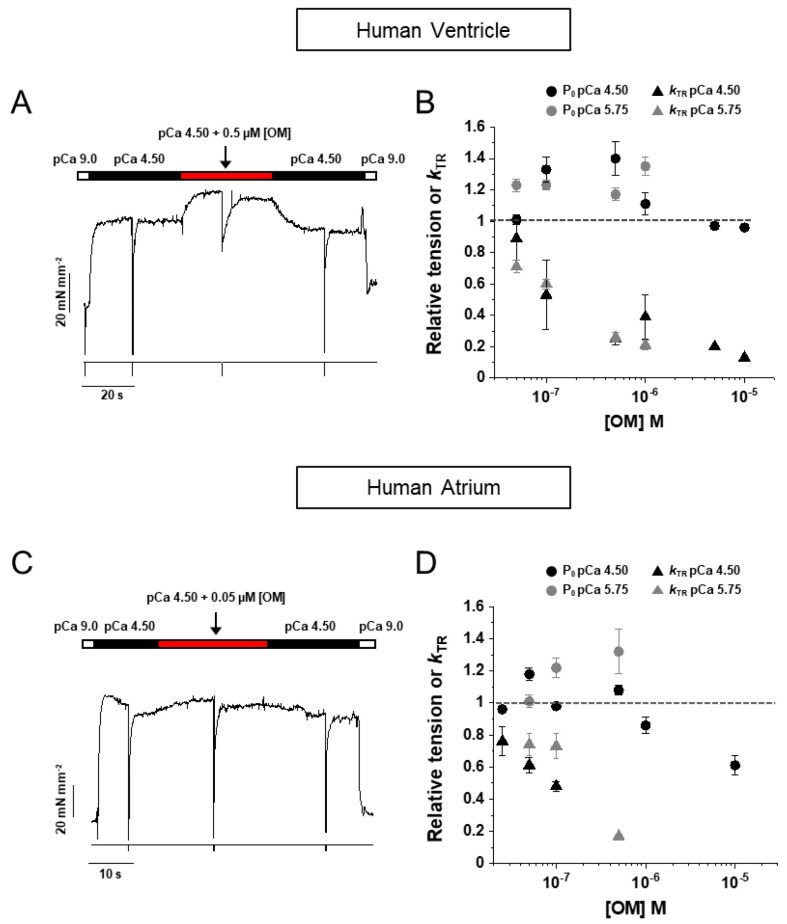
Effects of the rapid perturbation of [OM] during steady calcium-activation on tension development in human ventricular and atrial myofibrils (15 °C). (**A**): Representative trace from a jump experiment in aHV myofibril activated under control conditions and subjected to a rapid change in [OM] from 0 to 0.5 µM and back. The initial sarcomere length is 2.13 µm. The timing of the solution change is represented by the indexed bar at the top of the tension trace (top trace). Fast length changes (bottom trace) are applied to the myofibril under the condition of steady force generation (P_0_) under control and [OM] conditions to measure *k*_TR_. (**B**): Mean values of active tension P_0_ (dots) and *k*_TR_ triangles) measured from jump experiments (and normalised by the mean value of the same parameters measured before and after the ligand jump) plotted as a function of [OM]. Black symbols: jump experiments performed at maximal activation (pCa 4.50); grey symbols: jump experiments performed at submaximal activation (pCa 5.75). Error bars ± SEMs. (**C**): Representative trace from a jump experiment in a HA myofibril activated under control conditions and subjected to a rapid change in [OM] from 0 to 0.05 µM and back. The initial sarcomere length is 2.19 µm. The timing of the solution change is represented by the indexed bar at the top of the tension trace (top trace). Rapid changes in length (lower trace) are applied to the myofibril under conditions of steady force generation P_0_ in control and OM conditions to measure *k_TR_*. (**D**): Mean values of active tension P_0_ (dots) and *k*_TR_ (triangles) measured from jump experiments in HA myofibrils (and normalised by the mean value of the same parameters measured before and after the ligand jump) plotted as a function of [OM]. Black symbols: jump experiments performed at maximal activation (pCa 4.50); grey symbols: jump experiments performed at submaximal activation (pCa 5.75). Error bars± SEMs.

**Figure 3 ijms-25-09784-f003:**
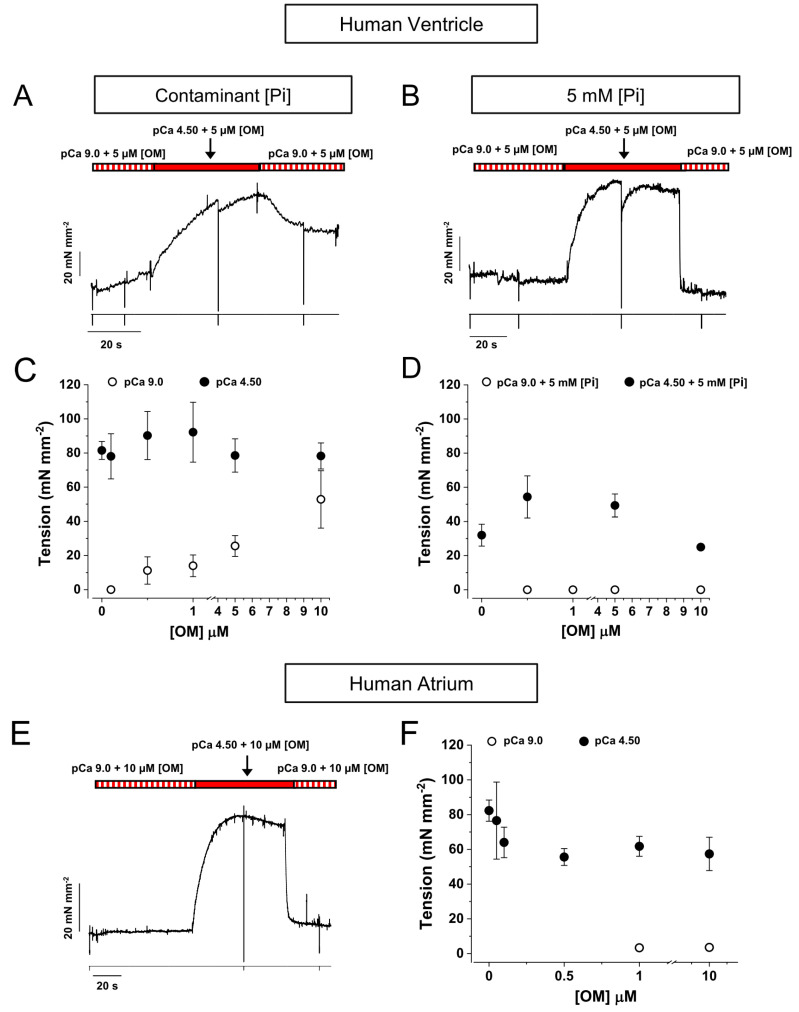
Effects of OM on the development of resting and calcium-activated tension in human ventricular and atrial myofibrils (15 °C). (**A**,**B**): Representative traces of tension generation in two human ventricular myofibrils activated and relaxed by fast solution change in the presence of 5 µM [OM] (A) or 5 µM [OM] and 5 mM [Pi] (**B**) in both relaxing and activating solutions; the timing of the solution change is represented by the indexed bars at the top of the tension traces (top traces); bottom traces: timing of rapid length changes applied to the myofibril to measure *k*_TR_. Sarcomere length: 2.03 µm (left panel) and 2.10 µm (right panel). (**C**,**D**): Mean absolute tension values measured in activationrelaxation cycles in the presence of different [OM]s (**C**) or in the presence of different [OM]s and 5 mM Pi (**D**). The [OM]s range from 0 to 10 µM. Filled symbols: active tension generation in pCa 4.50; empty symbols: Ca^2+^-independent force generation in relaxing solution (pCa 9.0) in the presence of [OM]. Error bars ± SEMs. (**E**): Representative trace of tension generation (top trace) in a HA myofibril activated and relaxed by fast solution change in the presence of 10 µM [OM] in both relaxing and activating solutions. The timing of the solution change is represented by the indexed bars at the top of the tension trace. Bottom trace: rapid length changes applied to the myofibril to measure *k*_TR_. Sarcomere length: 2.09 µm. (**F**): Mean absolute tension values measured in activation–relaxation cycles in control conditions and in the presence of different [OM]s up to 10 µM. Filled symbols: active tension generation in pCa 4.50; empty symbols: Ca^2+^-independent force generation in relaxing solution (pCa 9.0) in the presence of different [OM]s. Error bar, ±SEM.

**Figure 4 ijms-25-09784-f004:**
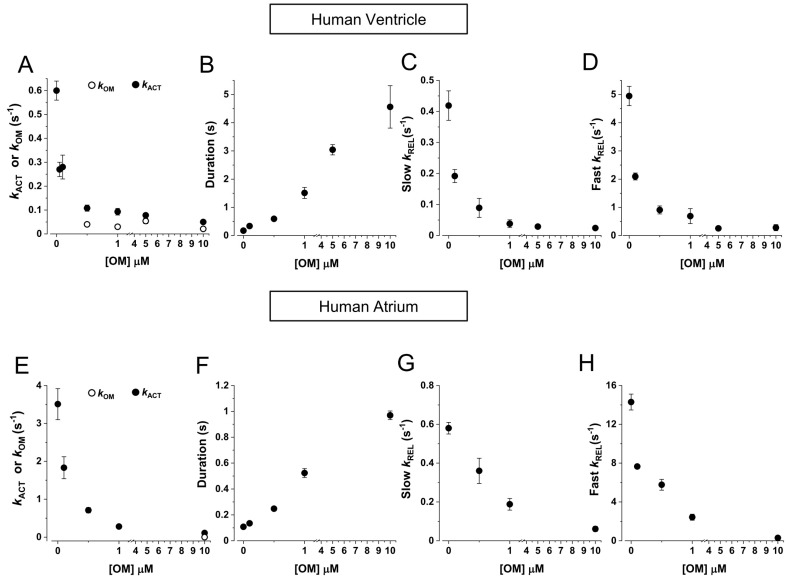
Effects of OM on the kinetics of tension generation and relaxation in human ventricular and atrial myofibrils (15 °C). (**A**–**D**): HV myofibrils. (**A**): Mean rate of tension generation (*k*_ACT_) at different [OM]s. Filled symbols: rate of tension development induced by switching from relaxing (pCa 9.0) to activating (pCa 4.5) solutions; empty symbols: *k*_OM_ rate of Ca^2+^-independent tension development induced by OM in relaxing solution (pCa 9.0). Error bars ± SEM. (**B**–**D**): Mean parameters of force relaxation in the presence of different [OM]s. (**B**): Duration of the slow phase of relaxation. (**C**): Rate of the slow phase of relaxation *slow k*_REL_. (**D**): Rate of the fast phase of relaxation *fast k*_REL_. (**E**–**H**): As above for HA myofibrils. (**E**): Mean values of *k*_ACT_ or *k*_OM_. (**F**): Duration of the slow phase of relaxation. (**G**): *slow k*_REL_ and H: *fast k*_REL_ in the presence of different [OM]s. Error bars ± SEM.

**Figure 5 ijms-25-09784-f005:**
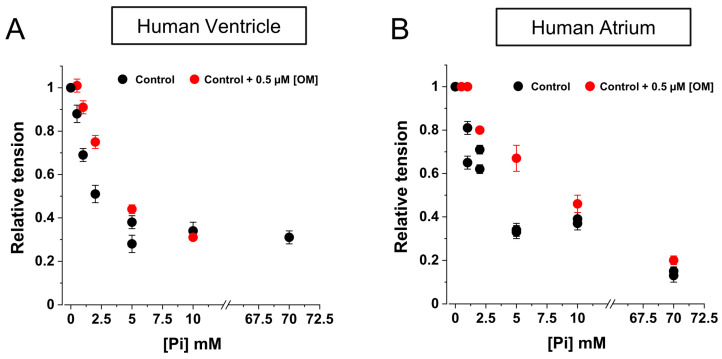
Effect of OM on the force/[Pi] relations of human ventricular and atrial myofibrils. Maximal Ca^2+^-activated isometric force in the presence of different [Pi] normalised over the force developed in the absence of added Pi under control conditions (black symbols) or in the presence of 0.5 µM [OM] (red symbols). (**A**): HV myofibrils; (**B**): HA myofibrils. Hyperbolic fitting of force/[Pi] relations in the absence of OM: Pi_50_, 0.83 ± 0.08 mM for HV and 2.12 ± 0.73 mM for HA myofibrils (asymptotes: 0.29 ± 0.01 and 0.15 ± 0.08, respectively).

**Figure 6 ijms-25-09784-f006:**
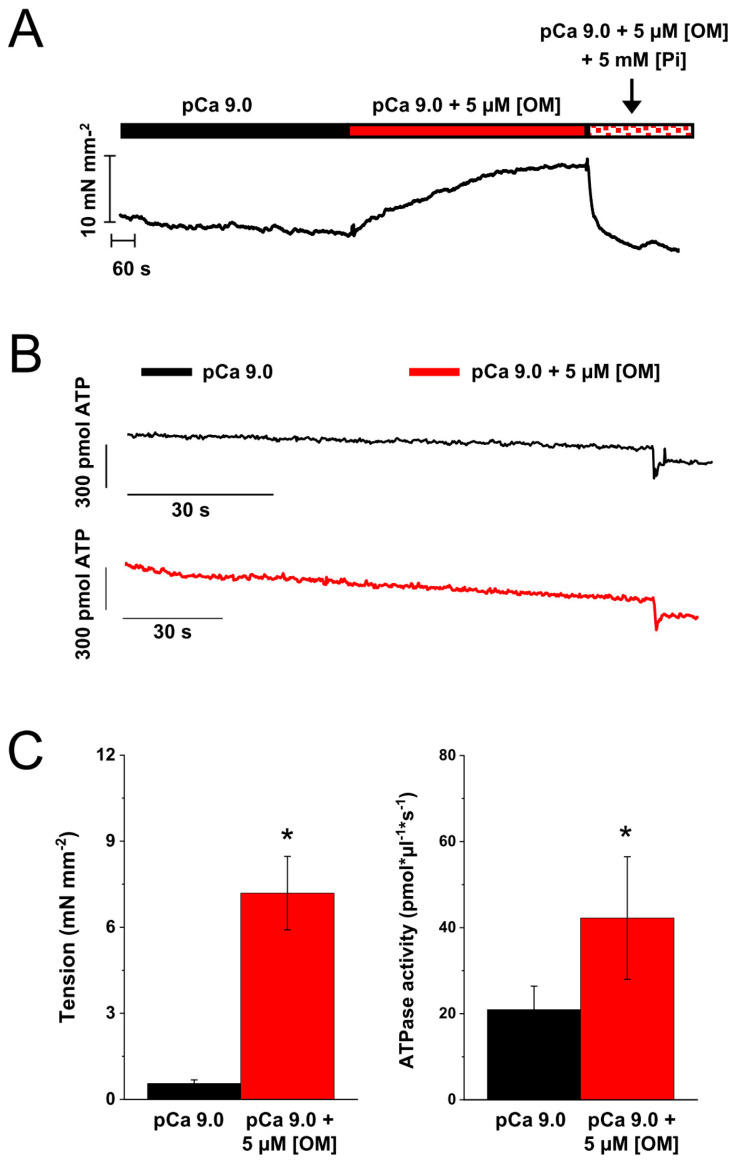
Effect of OM on the Ca^2+^-independent tension and the Ca^2+^-independent ATPase of skinned human left ventricular strips (25 °C). (**A**): Representative traces of resting tension in a HV skinned strips in pCa 9.0 or in pCa 9.0 added with 5 µM [OM] that causes Ca^2+^-independent force development. The timing of the solution change is represented by the indexed bar at the top of the tension trace. The arrow marks the time slot when the solution is changed to a pCa 9.0 solution added with both 5 µM [OM] and 5 mM [Pi]. The experiment shows that Pi abolishes Ca^2+^-independent/OM-induced force development. (**B**): ATPase measurements in pCa 9.0 (black) or in pCa 9.0 added with 5 µM [OM] (red) by enzyme coupled assay. (**C**): Mean values of Ca^2+^-independent tension (**left**) and Ca^2+^-independent ATPase (**right**) in control conditions (black bar) or in the presence of 5 µM [OM] (red bar). Error bars ± SEMs. Initial sarcomere length set to 2.2 µm. * *p* < 0.05, *n* = 5, paired *t* test.

**Table 1 ijms-25-09784-t001:** Mechanical and kinetic parameters of HV and HA myofibrils (15 °C, contaminant [Pi]).

Tension Generation			Relaxation		
			Slow Phase		Fast Phase
P_0_	*k* _ACT_	*k* _TR_	Duration	slow *k*_REL_	fast *k*_REL_
mN mm^−2^	s^−1^	s^−1^	ms	s^−1^	s^−1^
Human ventricle					
82 ± 5 (55)	0.60 ± 0.04 (23)	0.44 ± 0.01 (54)	173 ± 10 (23)	0.42 ± 0.05 (22)	4.95 ± 0.34 (23)
Human atrium					
82 ± 6 (49)	3.51 ± 0.41 (14)	2.96 ± 0.16 (44)	108 ± 6 (14)	0.58 ± 0.03 (14)	14.3 ± 0.82 (14)

Data are means ± SEMs; (*n*): number of myofibrils; P_0_: maximal Ca^2+^-activated tension; *k*_ACT:_ rate of tension development; *k*_TR_ rate of tension redevelopment; Duration: duration of the slow isometric phase of relaxation; *slow k*_REL_ and *fast k*_REL:_ rate of the slow and fast phases of relaxation, respectively.

## Data Availability

The data presented in this study are available upon request from the corresponding author.

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
