# Peer review of "Myosin Isoform-Dependent Effect of Omecamtiv Mecarbil on the Regulation of Force Generation in Human Cardiac Muscle"

_ijms, 2024, doi:10.3390/ijms25189784_

Round 1

Reviewer 1 Report

Comments and Suggestions for Authors

This study is investigating a small molecule, omecamtiv mecarbil (OM) that has been used in several clinical studies but is not approved by the FDA for use in HFrEF due to high mortality in patients with AF and side effects. According to the authors, “The main objectives of the study were to use OM as a sensitive tool to detect key differences in cross-bridge mechanical cycle in the presence of predominant α-fast or β-slow myosin heavy chain isoforms in relation to the regulatory state of ventricular and atrial sarcomeres, as well as to characterise the interplay between OM and Pi in modulating the mechanical performance of human myocardium”. The authors use human myofibrils from the atria and ventricle to determine the force under different conditions and in the absence and presence of inorganic phosphate.

Major concerns:

1.      The clinical relevance of this study is minimal. The authors confirm that OM might not be the best molecule to treat HFrEF because the effects are highly dependent on myosin composition of the myofibrils. Furthermore, the effects of OM are dependent on the [Ca2+] and presence of [Pi]. These are things that cannot be controlled in the human body. The basic conclusion of this study is that OM is a tool for detecting [Ca2+] and [Pi], as well as the distribution of fast and slow MyHC isoforms.

2.      The novelty and purpose of this study should be clarified. Other than possibly becoming a tool for detection, the authors do not present another use for OM. To take a molecule that has not lived up to the expectations in clinical trials and to investigate it further, one would hope that finding a reason for the failure and/or finding a way around that failure would benefit the therapeutic world more.

3.      The results subsections were organized in such a way that the reader is forced to look at as many as 3 different figures for one subsection of the results text. I would suggest either the figures or the text should be modified to be more cohesive.

Minor concerns:

1.      Figure 1a shows a western blot containing only 2 samples. Typically, a full blot is sent via supplemental data to show the entire gel. Or a blot with multiple lanes for each sample type should be included.

2.      Figure 5A and B contain additional symbols not included in the legend.

Comments on the Quality of English Language

The manuscript would benefit from being edited by a native English speaker. Many of the articles (a, an, the) are missing, and in some instances, the incorrect tense or word type (noun versus adverb) is used which changes the meaning of the sentence. The most challenging aspect of reading this manuscript was the use of run-on sentences. Many times, I was unable to follow the thought process until the end of the sentence and had to read the sentences multiple times to understand what the authors were trying to explain. 

Reviewer 2 Report

Comments and Suggestions for Authors

In their manuscript “Myosin isoform dependent effect of Omecamtiv Mecarbil on the regulation of force generation in human cardiac muscle” the authors aim to elucidate the complex mechanism underlying the action of this small molecule. For this, force generation is measured in ventricular and atrial myofibrils under various conditions including different concentrations of OM to analyze its dose-dependent effects as well as calcium and inorganic phosphate to draw conclusions about the molecular processes involved. In a concise study, the authors demonstrate significant differences in OM effects depending on the dominating myosin heavy chain isoforms in the myofibrils on the one hand and the relevance of phosphate for the OM mediated effects on the other hand. The experimental design is appropriate and the entire manuscript is well written. Overall, there is not much to object to from my point of view, so I will point out only minor issues in the following.

In the introduction, the last paragraph (page 2 line 95 – page 3 line 107) provides already a summary of all the major results presented in the next sections. As this goes far beyond an introduction, I would suggest simply to delete this paragraph here (the statements are repeated in the results and discussion anyway).

The methods section provides detailed information and implies rigid data curation even though reproducibility is naturally limited due to the usual experiment-related modificactions of the LabVIEW software. A minor point is that the authors did not mention age distribution and gender of the human donors, while especially the latter might be of interest. Sex-related differences in MHC isoform expression have been reported before (in mice: PMID 23179080 and in human: PMID 24878771), hence considering the gender when comparing the MyHC composition in atrial and ventricular myofibrils might reveal a further layer of complexity that could contribute to the inconsistent results in experimental studies and clinical trials.

Overall, the results are presented in a well-structured manner, however following their presentation in the results section at page 7 compels the reader to “jump” between the figures a lot. Potentially, this could be improved by rearranging the figure compositions in a way that follows the red line of the results more closely. Maybe, it might also be helpful to include the data for the tension in dependence of [OM] at pCa 4.5 (as illustrated in Fig 2B) in Fig. 2E to clearly demonstrate the point that Pi did not affect the “OM jump”. Unfortunately, I had some difficulties with the interpretation of Figure 5. For most Pi concentrations there are three data points (control, control+0.5µM OM and another symbol without legend entry), while for 70mM (Fig 5A) there is no data point for control+0.5µM OM, which left me a bit confused here. Accordingly, the caption should contain a more detailed explanation.

In contrast, the discussion section is very clear and the authors did a good job in thoroughly putting their results into context. The conclusion is reasonable and the results of the study could indeed prove valuable to the community.
